# Neuregulin-1 (NRG1) Binds to the Allosteric Binding Site (Site 2) and Suppresses Allosteric Integrin Activation by Inflammatory Cytokines: A Potential Mechanism of Anti-Inflammatory and Anti-Fibrosis Action of NRG1

**DOI:** 10.3390/cells14080617

**Published:** 2025-04-21

**Authors:** Yoko K. Takada, Yoshikazu Takada

**Affiliations:** 1Department of Dermatology, University of California School of Medicine, Research III Suite 3300, 4645 Second Ave., Sacramento, CA 95817, USA; 2Department of Biochemistry and Molecular Medicine, University of California School of Medicine, Research III Suite 3300, 4645 Second Ave., Sacramento, CA 95817, USA; 3VA Northern California Health Care System, 150 Muir Road, Martinez, CA 94553, USA

**Keywords:** neuregulin-1, integrin, allosteric activation

## Abstract

We showed that multiple inflammatory cytokines (e.g., CCL5, CXCL12, CX3CL1, CD40L, and FGF2) bind to the allosteric site (site 2) of integrins, distinct from the classical RGD-binding site (site 1), and allosterically activate integrins. A major inflammatory lipid mediator 25-hydroxycholesterol is known to bind to site 2 and allosterically activates integrins and induces inflammatory signals (e.g., IL-6 and TNF secretion). Thus, site 2 is involved in inflammatory signaling. Neuregulin-1 (NRG1) is known to suppresses the progression of inflammatory diseases, fibrosis, and insulin resistance. But, the mechanism of anti-inflammatory action of NRG1 is unclear. We previously showed that NRG1 binds to the classical RGD-binding site (site 1). Mutating the 3 Lys residues that are involved in site 1 binding (NRG1 3KE mutant) is defective in binding to site 1 and in ErbB3-mediated mitogenic signals. Docking simulation predicted that NRG1 binds to site 2. We hypothesized that NRG1 acts as an antagonist of site 2 and blocks allosteric activation by multiple cytokines. Here, we describe that NRG1 binds to site 2 but does not activate soluble αvβ3 or αIIbβ3 in 1 mM Ca^2+^, unlike inflammatory cytokines. Instead, NRG1 suppressed integrin activation by several inflammatory cytokines, suggesting that NRG1 acts as a competitive inhibitor of site 2. Wild-type NRG1 is not suitable for long-term treatment due to its mitogenicity. We showed that the non-mitogenic NRG1 3KE mutant still bound to site 2 and inhibited allosteric activation of soluble and cell-surface integrins, suggesting that NRG1 3KE may have potential as a therapeutic.

## 1. Introduction

Integrins are a superfamily of α-β heterodimers of cell adhesion receptors and recognize extracellular matrix (ECM) proteins, cell-surface molecules (e.g., ICAM-1 and VCAM-1), and small proteins (e.g., cytokines) [1]. The integrin ligands are known to bind to the classical RGD-binding site (site 1) in the headpiece. Integrins are involved in cytokine signaling (integrin–cytokine crosstalk) since antagonists to integrins suppress cytokine signaling [2,3,4]. Previous studies showed that several cytokines such as FGF1 [5,6], IGF1 [7,8], and neuregulin-1 (NRG1) [9] induce the integrin–cytokine–cognate receptor ternary complex on the surface (Figure 1). Cytokine mutants defective in integrin binding were defective in signaling functions and acted as antagonists of cytokine signaling [5,8,9]. We showed that pro-inflammatory sPLA2-IIA [10] and CD40L [11] bind to site 1.

It has been proposed that signals from inside the cells (inside-out signaling) mediates integrin activation [12,13]. However, we recently showed that multiple inflammatory cytokines bind to an allosteric ligand binding site of integrins (site 2) [14,15,16,17,18], which is on the opposite side of site 1 in the integrin headpiece. The binding of inflammatory cytokines (e.g., CCL5) to site 2 activates integrins in the absence of canonical inside-out signaling (allosteric activation). We also showed that sPLA2-IIA [15] and CD40L [17,19] bind to site 2 and activate integrins. The biological role of site 2 has not been fully identified. A recent study showed that 25-hydroxycholesterol, a major lipid pro-inflammatory mediator, has been shown to bind to site 2, activate integrins, and induce pro-inflammatory signaling (e.g., IL-6 and TNF secretion) [20] (Figure 1). Therefore, it is likely that site 2 mediates pro-inflammatory signaling by inflammatory cytokines and inflammatory mediators. Thus, the binding of inflammatory cytokines and inflammatory mediators to site 2 may represent a common mechanism of inflammatory signaling.

FGF1 is anti-inflammatory and prevents the development of several inflammatory diseases [21]. Also, FGF1 lowers blood glucose levels in diabetic mice [22]. FGF1 is also cardioprotective (anti-thrombotic), and blocking FGF1 synthesis by activating FGF1 promoter methylation exacerbated deep vein thrombosis [23]. However, the mechanism of the anti-inflammatory and cardioprotective action of FGF1 has not been established. We discovered that FGF1 bound to site 2 but did not activate integrins, and, in contrast, pro-inflammatory FGF2 bound to site 2 and activated integrins [24]. Notably, FGF1 inhibited integrin activation induced by FGF2 [24]. Thus, it is likely that FGF1 is a natural antagonist for site 2-mediated inflammatory signaling.

The neuregulins are a family of four structurally related proteins that are part of the EGF family (NRG1–4). The extracellular domain that contains an EGF-like motif that binds and activates receptor tyrosine kinases in the EGF receptor (ErbBs) family is released by cleavage by metalloproteases from transmembrane NRGs, which typically function as precursor molecules. The extracellular domain may subsequently bind to nearby receptors (autocrine/paracrine action). NRG1 binds to ErbB3 and ErbB4. Like several cytokines described above, we showed that NRG1 binds to site 1 of integrin αvβ3 and induces integrin–NRG1–ErbB3 ternary complex formation, which is critical for its mitogenic action. The NRG1 mutant defective in binding to site 1 (3KE) is defective in inducing ErbB3 activation and acts as an antagonist for ErbB3 [9].

Notably, NRG1 is known to be neuroprotective [25]. Earlier studies showed loss of NRG1 expression in multiple sclerosis-active lesions [26]. The expression of NRG1 decreased in lysolecithin-induced focal demyelinating lesions [27]. The EGF-like domain of NRG1 was shown to suppress experimental autoimmune encephalomyelitis (EAE), an animal model of multiple sclerosis [25,28]. NRG1 levels reduced in multiple sclerosis, and exogenous NRG1 reduced the disease progression in animal models [25]. Also, the levels of pro-inflammatory chemokines (e.g., CCL2, CCL5) are increased in neural tissues in multiple sclerosis and are reduced after treatments [29]. It has been proposed that NRG1 inhibits the release of inflammatory factors in microglia and regulates the switching of the M1/M2 microglia phenotype [30]. Chronic NRG1 treatment is known to suppress myocardial apoptosis and fibrosis after myocardial infarction [31]. It is also known that NRG1 lowered blood glucose and improved insulin sensitivity [32]. However, the mechanisms of these anti-inflammatory, anti-fibrosis, and glucose-lowering actions of NRG1 have not been fully established. We hypothesized that NRG1 binds to site 2 and blocks inflammatory signaling induced by inflammatory cytokines.

In the present paper, we show that docking simulation predicted that NRG1 (the EGF-like domain) binds to site 2. NRG1 did not allosterically activate soluble integrins unlike several inflammatory cytokines and instead competitively suppressed integrin activation by multiple pro-inflammatory cytokines. We thus propose that NRG1 suppresses the binding of pro-inflammatory cytokines to site 2 and subsequent pro-inflammatory signaling, as predicted. We propose that blocking site 2-mediated inflammatory signaling may be a common mechanism of anti-inflammatory action by NRG1 and FGF1.

## 2. Materials and Methods

### 2.1. Protein Expression

The truncated fibrinogen γ-chain C-terminal domain (γC399tr) was generated as previously described [33]. Fibrinogen γ-chain C-terminal residues 390–411 cDNA encoding (6 His tagged) [HHHHHH]NRLTIGEGQQHHLGGAKQAGDV] was conjugated with the C-terminus of GST (designated γC390-411) in pGEXT2 vector (BamHI/EcoRI site). The protein was synthesized in *E. coli* BL21 and purified using glutathione affinity chromatography. FGF2 was synthesized as previously described [34].

The WT NRG1 EGF-like domain [GTSHLVKCAEKEKTFCVNGGECFMVKDLSNPSRYLCKCQPGFTGARCTENVPMKVQNQEKAEELYQK] and its 3KE mutant were synthesized in endotoxin-free *E. coli* BL21 (DE3) pLysS Clear-coli by inducing with 0.2 mM IPTG for 3 h at 37 °C. His-tagged NRG1 was purified by Ni-NTA affinity chromatography from bacterial extracts in denaturing conditions in 8M urea. The purified His-tagged NRG1 preparations were re-folded as described [7].

The cDNA fragment encoding the specificity-determining loop (SDL) (residues 158–188 of β3 [DKPVSPYMYISPPEALENPCYDMKTTCLPMF]) was synthesized and subcloned into the Bam HI/Eco RI site of pGEX2T vector. Proteins were synthesized in *E. coli* BL21 and purified using Glutathione Sepharose affinity chromatography.

We synthesized the 29 mer cyclic β3 site 2 peptide C260- RLAGIV[QPNDGSHVGSDNHYSASTTM]C288 (C273 is changed to S273) by inserting oligonucleotides encoding this sequence into the BamHI/EcoRI site of pGEX2T vector. We selected the positions of Cys residues for disulfide linkage by using Disulfide by Design-2 (DbD2) software v2.12 (http://cptweb.cpt.wayne.edu/DbD2/ (accessed on 13 April 2020)) [35]. It is predicted that mutating Gly260 and Asp288 to Cys disulfide-linked cyclic site 2 peptide of β3 does not affect the conformation of the original site 2 peptide sequence QPNDGSHVGSDNHYSASTTM in the 3D structure. We found that the cyclic site 2 peptide bound to CX3CL1 and sPLA2-IIA to a similar extent to non-cyclized β3 site 2 peptides in ELISA-type assays. We designed the corresponding cyclic β1 peptide (C268KLGGIVLPNDGQSHLENNMYTMSHYYC295, 28- mer cyclic β1 peptide), in which C281 is converted to S281. We synthesized the proteins in BL21 cells and purified using Glutathione Sepharose affinity chromatography.

Site-directed mutagenesis was performed using the QuikChange method [36]. We verified the presence of the mutations by DNA sequencing.

### 2.2. Docking Simulation

We performed docking simulation of the interaction between NRG1 and integrin αvβ3 (closed headpiece form, PDB code 1JV2) using AutoDock3.05, as described previously [9]. The headpiece (residues 1–438 of αv and residues 55–432 of β3) of αvβ3 was used. Integrins do not contain cations during docking simulation, as in the previous studies using αvβ3 (open headpiece form, 1L5G.pdb) [5].

### 2.3. Statistical Analyses

We used Prism 10 (Graphpad Software, Boston, MA, USA) to test treatment differences using ANOVA and Tukey multiple comparison tests to control the global type I error.

We performed activation of soluble β3 integrins using ELISA-type activation assays as described previously [15,37]. Briefly, we coated wells of 96-well Immulon 2 microtiter plates (Dynatech Laboratories, Chantilly, VA, USA) with 100 µL 0.1 M PBS containing γC390-411 for αIIbβ3 and γC399tr for αvβ3 for 2 h at 37 °C. We blocked the remaining protein-binding sites by incubating with PBS/0.1% BSA for 30 min at room temperature. After washing with PBS, soluble recombinant β3 integrins (1 µg/mL) in the presence or absence of NRG1 and/or inflammatory cytokines were added to the wells and incubated in HEPES–Tyrodes buffer (10 mM HEPES, 150 mM NaCl, 12 mM NaHCO_3_, 0.4 mM NaH_2_PO_4_, 2.5 mM KCl, 0.1% glucose, 0.1% BSA) with 1 mM CaCl_2_ for 1 h at room temperature. After unbound β3 integrins were removed by rinsing the wells with binding buffer, bound β3 integrins were measured using anti-integrin β3 mAb (AV-10) followed by HRP-conjugated goat anti-mouse IgG and peroxidase substrates.

### 2.4. Site 2 and SDL Peptide Binding to NRG1

Wells of 96-well microtiter plate were incubated with NRG1 (20 μg/mL in PBS) for 1 h at room temperature, and the remaining protein-binding sites were blocked with BSA (0.1%). Wells were incubated with peptides fused to GST in PBS/Tween20 (0.05%) for 1 h at room temperature. Wells were washed with PBS/Tween20, and bound peptides were measured using HRP-conjugated anti-GST antibody. Signals with scrambled β3 site 2 peptide as a negative control were subtracted.

### 2.5. Blocking Activation of Cell-Surface αIIbβ3 by NRG1 3KE

We cultured αIIbβ3-CHO cells (the cell line is generated in our lab) in DMEM/10% FCS [18]. We resuspended cells with HEPES–Tyrodes buffer/0.02% BSA (heat-treated at 80 °C for 20 min to remove contaminating cell adhesion molecules). We incubated the αIIbβ3-CHO and then incubated with chemokines for 30 min on ice with or without NRG1 3KE and then incubated with FITC-labeled γC390-411 (50 μg/mL) for 60 min at room temperature. We washed the cells with PBS/0.02% BSA and analyzed in BD Accuri flow cytometer (Becton Dickinson, Mountain View, CA, USA). We analyzed the data using FlowJo 7.6.5, BD Bioscience, Franklin Lakes, NJ, USA.

## 3. Results

### 3.1. Docking Simulation Predicts That NRG1 Binds to Site 2

Since anti-inflammatory FGF1 binds to site 2 and suppresses integrin activation by pro-inflammatory FGF2, we hypothesized that NRG1 binds to site 2 and suppresses binding of pro-inflammatory cytokines to site 2 [24]. Closed headpiece structure of αvβ3 (1JV2.pdb) is well-defined in αvβ3 and has been successfully used for ligand binding to site 2. We thus performed docking simulation of the interaction between NRG1 and integrin αvβ3 (1JV2.pdb). The first cluster of docking poses with docking energy −24 kcal/mol represent the most likely poses of NRG1 EGF-like domain that bind to site 2 (Figure 2a,b). The docking energy is the one of the lowest among several other proteins that bind to site 2 (e.g., CCL5, FGF1) [18,24], suggesting that NRG1 is among the site 2-binding proteins with highest affinity. A previous study showed that the Lys residues at positions 5, 9, and 11 in the site 1 binding interface are critical for site 1 binding [9]. These Lys residues are not involved in site 2 binding, predicting that site 2-binding interface is distinct from that of site 1. Thus, mutation of these Lys residues (3KE mutation) suppresses the NRG1 binding to site 1 but does not affect binding to site 2, as shown later.

### 3.2. Site 2-Derived Peptide and Specificity-Determining Loop (SDL) Peptide Bind to NRG1

Previous studies showed that peptides from the binding site (site 2 peptide) bound to ligands [15,16,17,18,19,24,37]. We found that cyclic site 2 peptides weakly bound to NRG1 (Figure 2c). We noticed that another group of amino acid sequences in site 2 were involved in NRG1 binding to site 2 (Table 1). We found that the peptide (β3 SDL peptide) fused to GST bound to NRG1 better than site 2 peptides (Figure 2c). Site 2 peptides and the SDL peptide are close to the NRG1 EGF in the docking model (Figure 2d). These findings are consistent with the prediction that an NRG1 EGF-like domain binds to site 2. In our previous study [9], the predicted NRG1 binding interface to site 1 (nine amino acid residues in β3) was listed (Table 1 [9]). In the present study, we listed amino acid residues involved in site 2 binding in β3 (Table 1). We found that only a single amino acid residue (Glu171) is common to site 1 and site 2, and the remaining 14 out of 15 amino acid residues are only involved in NRG1-binding interface binding in site 2. This suggests that SDL is almost completely part of site 2 and is useful to identify site 2-binding proteins.

We previously showed that NRG1 binds to site 1. To define the positions of site 1 and site 2, we show the docking models of (a) binding of NRG1 to site 1 [9], (b) NRG1 binding to site 2 (the present study), and (c) in which the two models are superposed (Figure 3). The αvβ3 is present in two different conformations, although they are superposable: open headed/active (1L5G.pdb) or closed headed/inactive (1JV2.pdb).

### 3.3. NRG1 Does Not Activate β3 Integrins and Instead Suppresses Activation Induced by Inflammatory Cytokines

All the cytokines that are predicted to bind to site 2 allosterically activated except for FGF1 [24], and therefore we expected that NRG1 EGF will activate integrins. We tested if NRG1 suppresses allosteric integrin activation by several inflammatory cytokines. In the ELISA-type activation assays, we measured if NRG1 increases the binding of soluble β3 integrins to the immobilized fibrinogen fragments (γC399tr specific for αvβ3 or γC390-411 specific for αIIbβ3). Soluble β3 integrins are not active in 1 mM Ca^2+^ in the absence of inflammatory cytokines. Unexpectedly, we found that NRG1 did not activate soluble β3 integrins in ELISA-type activation assays, although FGF2 as a positive control activated them (Figure 4a,b). We studied if NRG1 EGF suppressed allosteric activation by four inflammatory cytokines that allosterically activate integrins. We found that NRG1 suppressed activation of soluble αvβ3 integrin by FGF2, CCL5, CX3CL1, and CXCL12 in a dose-dependent manner (Figure 5). We obtained similar results using soluble αIIbβ3 (Figure 6). These findings suggest that NRG1 binds to site 2 but acts as a competitive inhibitor of allosteric integrin activation induced by several inflammatory cytokines.

### 3.4. Non-Mitogenic NRG1 3KE Mutant Suppresses Integrin Activation Induced by Inflammatory Cytokines Through Site 2 Binding

NRG1 is a potent mitogen and cannot be used for long-term treatment since it activates ErbB3 and ErbB4. We previously reported that the 3KE mutation in the NRG1 EGF blocked NRG1 binding to integrin site 1 and suppressed integrin–NRG1–ErbB3 ternary complex formation and subsequent ErbB3 activation [9]. The docking simulation predicts that the 3KE mutation is not involved in NRG1 binding to site 2 (Figure 2a), suggesting that NRG1 3KE can bind to site 2. We studied whether the non-mitogenic NRG1 3KE mutant can suppress integrin activation induced by inflammatory CCL5 chemokine. Notably, the NRG1 3KE suppressed integrin activation by CCL5 (Figure 7a,b). These findings suggest that the NRG1 3KE can be used as a non-mitogenic anti-inflammatory agent. These findings also suggest that NRG1 acts independent of ErbB3. It should be noted that allosteric activation of integrins by multiple inflammatory cytokines and its suppression by NRG1 occur independent of cognate receptors (e.g., receptor tyrosine kinases and G-protein-coupled receptors).

It is still unclear whether NRG1 binds to site 2 in more biological conditions when integrins are expressed on the cell surface. We previously showed that cell-surface αIIbβ3 on CHO cells (αIIbβ3-CHO) is activated by inflammatory cytokines [18]. We found that excess NRG1-3KE effectively suppressed allosteric integrin activation induced by three inflammatory cytokines (Figure 8). These findings suggest that NRG1 3KE suppresses allosteric activation of integrins on the cell surface, and this inhibition may be a potential anti-thrombotic effect by NRG1 and is biologically relevant.

## 4. Discussion

A major finding in the present study is that NRG1 binds to site 2 and inhibits allosteric activation of integrin αvβ3 and αIIbβ3 by several inflammatory cytokines, suggesting that NRG1 is a competitive inhibitor of site 2-mediated allosteric activation by several inflammatory cytokines. Activation of αIIbβ3 is a key event that leads to platelet aggregation and thrombosis. We previously showed that inflammatory cytokines bind to site 2 of αIIbβ3 and activate this integrin (see Section 1), leading to platelet activation and aggregation. It is unknown if NRG1 suppresses platelet activation. Also, allosteric activation of αvβ3 by inflammatory cytokines may be related to pro-inflammatory and pro-fibrosis action of inflammatory cytokines. In the present study, we showed that NRG1 suppressed allosteric activation of vascular integrin αvβ3. This is a potential mechanism of anti-inflammatory, anti-fibrosis, and anti-insulin-resistance actions of NRG1. Furthermore, we showed that inhibition of allosteric integrin activation occurs in cell-surface integrin αIIbβ3, suggesting that this inhibition is biologically relevant. We suspect that inflammatory cytokines allosterically activate multiple integrins other than αvβ3 and αIIbβ3, and NRG1 inhibits this activation.

We superposed the multiple docking models of inflammatory cytokines that bind to site 2 (e.g., CCL5, CX3CL1, CXCL12, and FGF2) and NRG1 with αvβ3 headpiece (1 JV2) (Figure 9). Previous studies showed the predicted binding sites of multiple inflammatory cytokines in site 2. Inflammatory cytokines are clustered in the center of site 2 (Figure 9). But, the predicted NRG1 binding site is off the center of site 2. One possible mechanism of anti-inflammatory action of NRG1 would be that NRG1 binds to site 2 at high affinity, but its binding site is off the center of site 2, and thereby NRG1 cannot induce allosteric integrin activation. NRG1 EGF inhibits allosteric activation by inflammatory cytokines by blocking the binding of inflammatory cytokines to the center of site 2 (Figure 9). These binding models predict how NRG1 acts as an antagonist of site 2. Notably, we showed that non-mitogenic NRG1 3KE suppressed allosteric activation of integrins by CCL5. It is unlikely that cognate receptors are involved in the glucose-lowering action of NRG1.

We used very high concentrations (up to 100 μg/mL) of cytokines and NRG1 in our activation and inhibition experiments using soluble integrins in cell-free conditions. We detected allosteric activation of cell-surface integrins and inhibition by NRG1 3KE at much lower concentrations (Figure 8). It is likely that they are expected to interact with integrins at much lower concentrations on the cell surface, since cytokines and NRG1 bind to cell-surface proteoglycans or membrane-bound ones (e.g., CX3CL1 and NRG1), and therefore they are expected to be highly concentrated on the cell surface.

## Figures and Tables

**Figure 1 cells-14-00617-f001:**
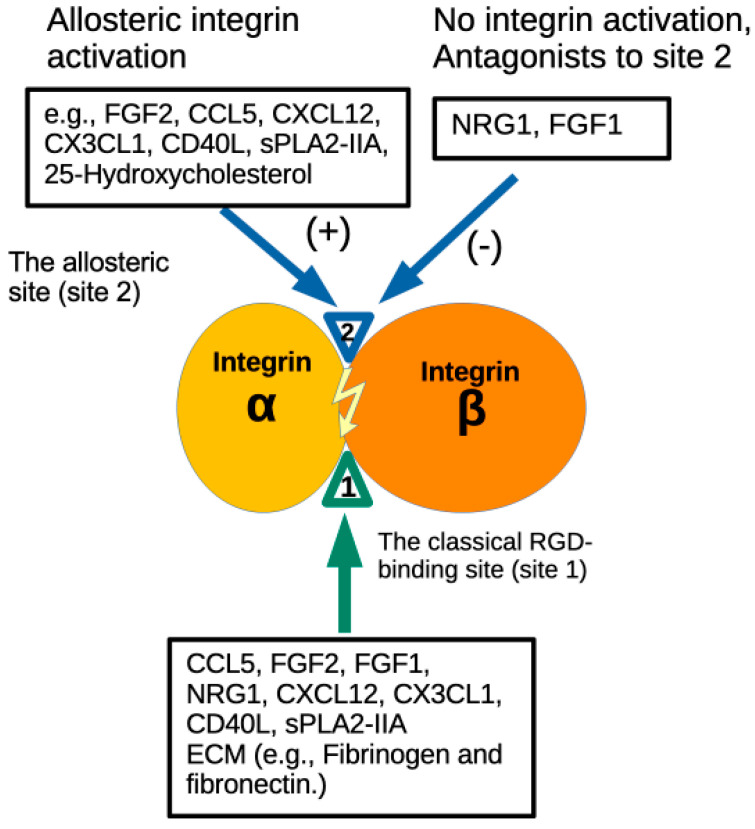
A model of regulation of integrin activation and inflammatory signals by pro- and anti-inflammatory cytokines: Several pro-inflammatory cytokines and 25-hydroxycholesterol (25HC), a mediator of inflammatory signals, bind to site 2 (in addition to site 1) and induce integrin activation and pro-inflammatory signals. Anti-inflammatory FGF1 binds to site 1 and site 2 but does not activate integrins or induce pro-inflammatory signals.

**Figure 2 cells-14-00617-f002:**
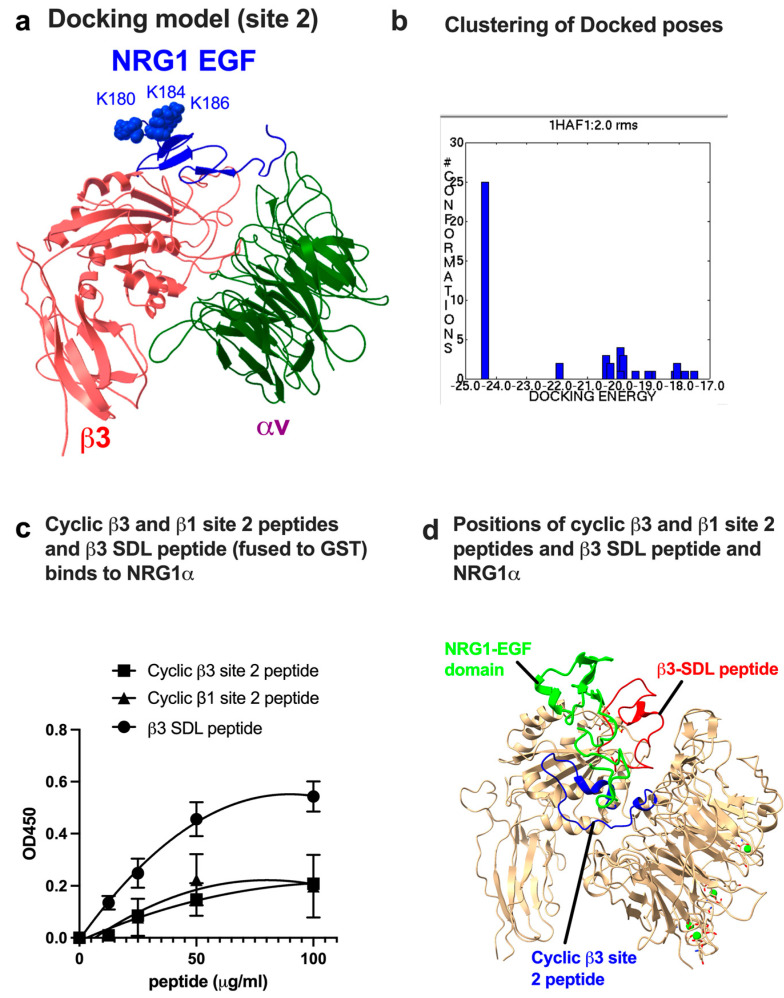
NRG1 EGF is predicted to bind to site 2. (**a**) Docking simulation using closed headpiece of αvβ3 (1JV2.pdb) predicts that NRG1 binding affinity to site 2 is high (docking energy -24.3 kcal/mol). Amino acid residues involved in αvβ3-NRG1 interaction are described in Table 1. (**b**) Clustering of docking poses. The first cluster represents the optimum docking poses for NRG1 binding to site 2. (**c**) NRG1 EGF domain binds to cyclic site 2 peptide and β3 SDL peptide. NRG1 (20 μg/mL) was immobilized to wells and incubated with peptides fused to GST in PBS + Tween20 (0.05%) for 1 h at room temperature. Bound peptides were measured using HRP-conjugated anti-GST antibody. Signals with scrambled β3 site 2 peptide as a negative control were subtracted. (**d**) Positions of cyclic site 2 peptide and β3-SDL peptide.

**Figure 3 cells-14-00617-f003:**
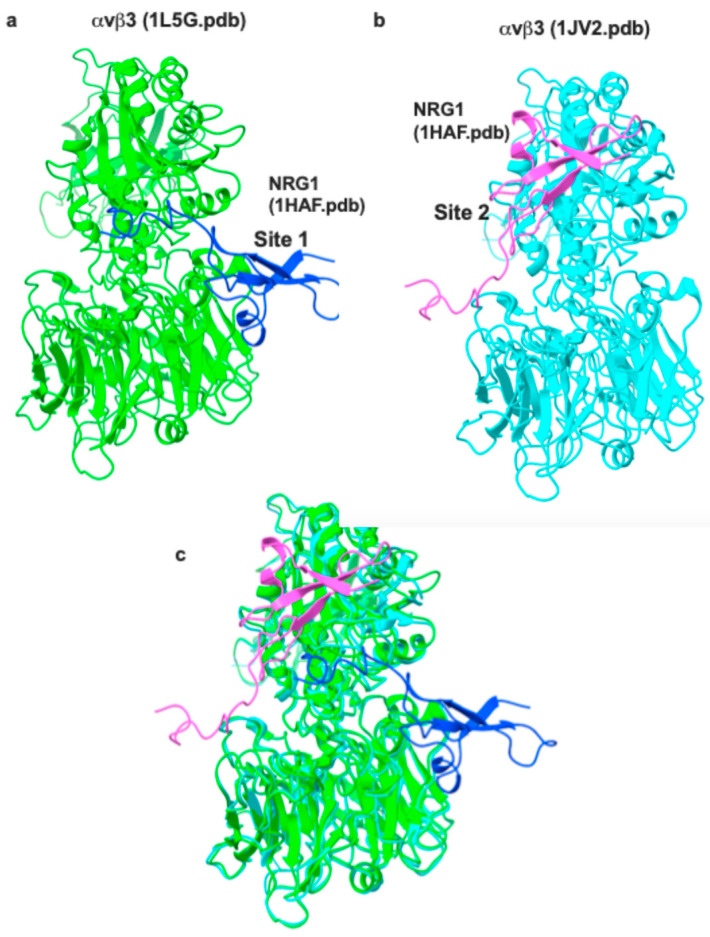
Docking models in which NRG1 binds to two distinct sites of αvβ3 (site 1 and site 2). When integrin αvβ3 is activated (e.g., by 1 mM Mn^2+^ or Mg^2+^), it takes open headed/active conformation (1L5G.pdb), and site 1 (the classical RGD-binding site) is open. When 1L5G.pdb is used for docking simulation, NRG1 is predicted to bind to site 1 (**a**) [9]. When αvβ3 is inactive (in closed headpiece, 1JV2.pdb) (e.g., in body fluid with high [Ca^2+^]), NRG1 is predicted to bind to site 2 (**b**). Inactive (**a**) and active (**b**) forms of αvβ3 were superposed in (**c**) to show site 1 and site 2 are distinct.

**Figure 4 cells-14-00617-f004:**
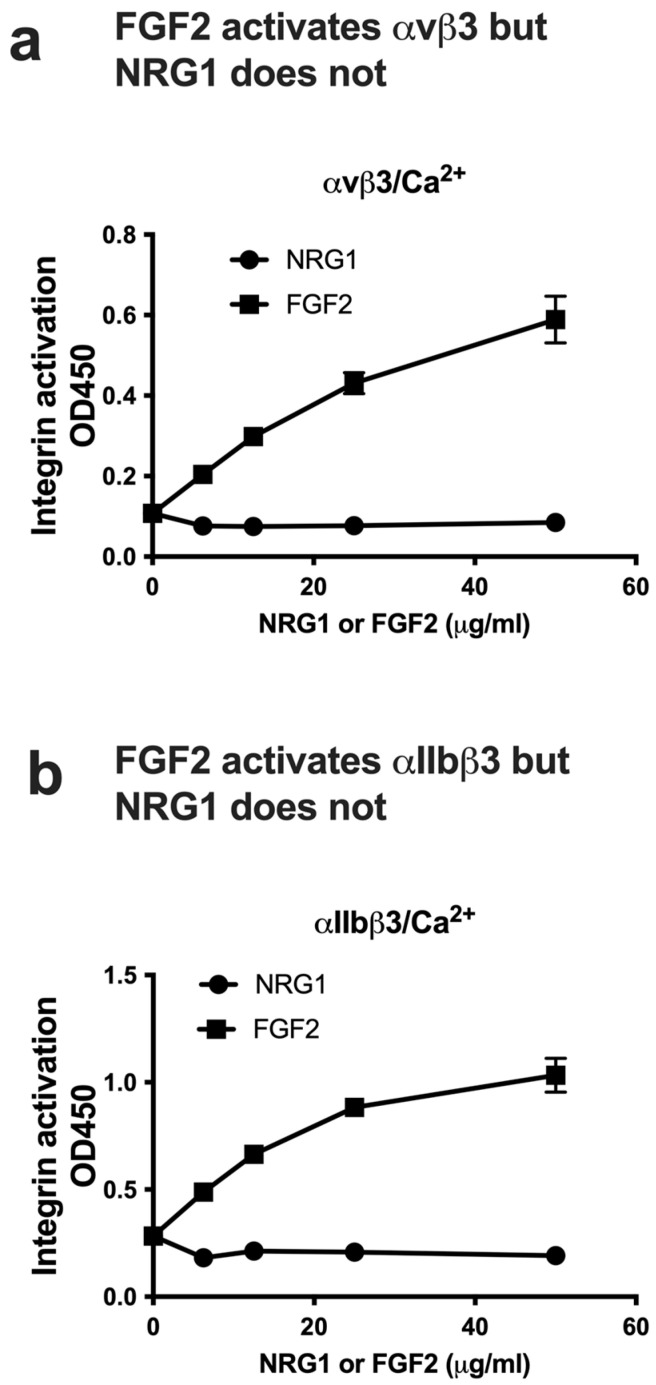
NRG1 did not activate soluble αvβ3 and αIIbβ3 in 1 mM Ca^2+^, but FGF2 did. (**a**) NRG1 did not activate αvβ3, although FGF2 (as a positive control) did in ELISA-type activation assays. Wells of 96-well microtiter plate were coated with the C-terminal domain of fibrinogen γ-chain, in which the C-terminal 12 residues are truncated (γC399tr), a specific ligand for integrin αvβ3 (50 μg/mL). Wells were incubated with soluble integrin αvβ3 (1 μg/mL) in Tyrode–HEPES buffer with 1 mM Ca^2+^ for 1 h at room temperature in the presence of NRG1 or FGF2, and bound integrins were quantified using anti-β3 mAb and anti-mouse IgG. Data are shown as average ± SD in triplicate experiments. (**b**) NRG1 did not activate αIIbβ3, although FGF2 (as a positive control) did in ELISA-type activation assays. Activation assays were performed as described in (**a**), except that wells were coated with the C-terminal domain of fibrinogen γ-chain C-terminal 12 residues (γC390-411), a specific ligand for integrin αIIbβ3 (20 μg/mL), and incubated with soluble integrin αIIbβ3 (1 μg/mL).

**Figure 5 cells-14-00617-f005:**
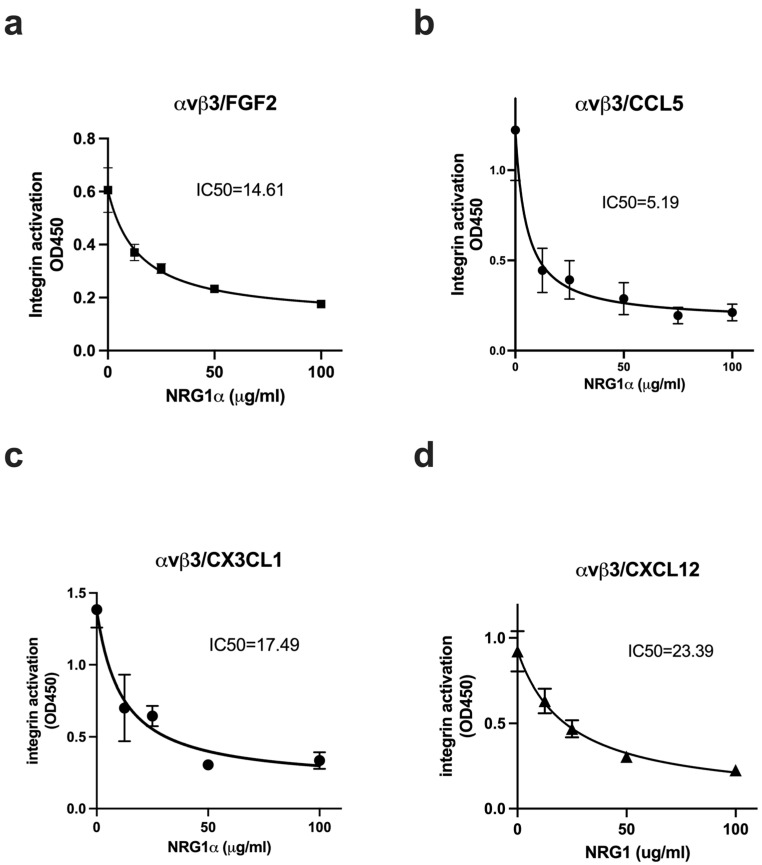
NRG1 suppressed allosteric activation of soluble αvβ3 by multiple inflammatory cytokines. Wells of 96-well microtiter plate were coated with the C-terminal domain of fibrinogen γ-chain, in which the C-terminal 12 residues are truncated (γC399tr), a specific ligand for integrin αvβ3 (50 μg/mL). Soluble αvβ3 was activated by (**a**) FGF2 (12.5 μg/mL), (**b**) CCL5 (at 6.25 μg/mL), (**c**) CX3CL1 (at 6.25 μg/mL), and (**d**) CXCL12 (at 6.25 μg/mL) in the presence of NRG1 EGF domain. Data are shown as average ± SD in triplicate experiments.

**Figure 6 cells-14-00617-f006:**
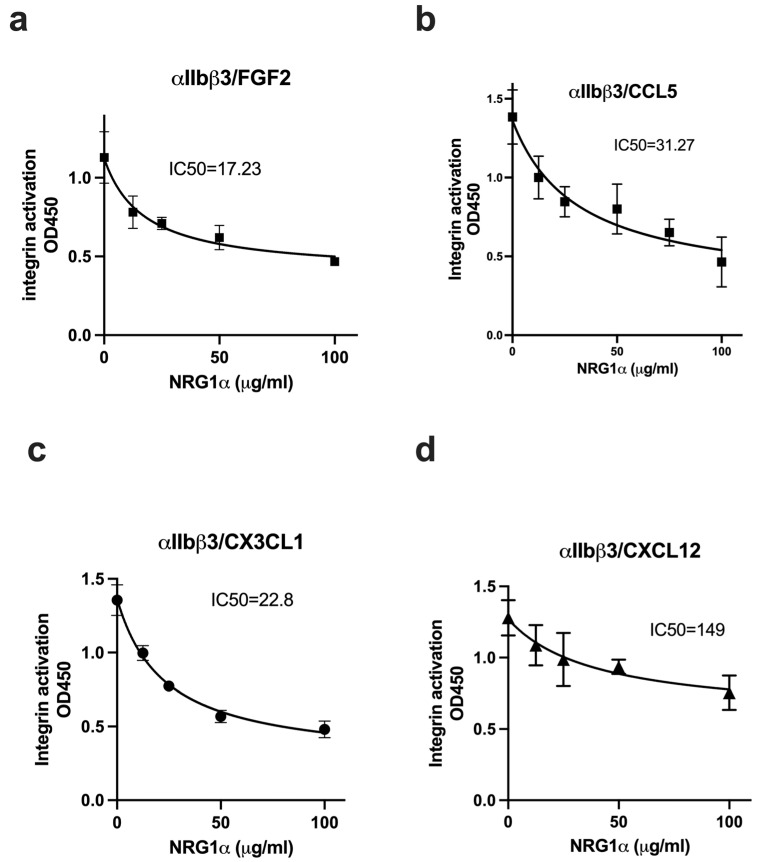
NRG1 suppressed allosteric activation of soluble αIIbβ3 by multiple inflammatory cytokines. Wells of 96-well microtiter plate were coated with the C-terminal domain of fibrinogen γ-chain C-terminal 12 residues (γC390-411), a specific ligand for integrin αIIbβ3 (20 μg/mL). Soluble αIIbβ3 (1 μg/mL) was activated by (**a**) FGF2 (12.5 μg/mL), (**b**) CCL5 (at 6.25 μg/mL), (**c**) CX3CL1 (at 6.25 μg/mL), and (**d**) CXCL12 (at 6.25 μg/mL) in the presence of NRG1. Background binding in the absence of cytokines was not subtracted. Data are shown as average ± SD in triplicate experiments.

**Figure 7 cells-14-00617-f007:**
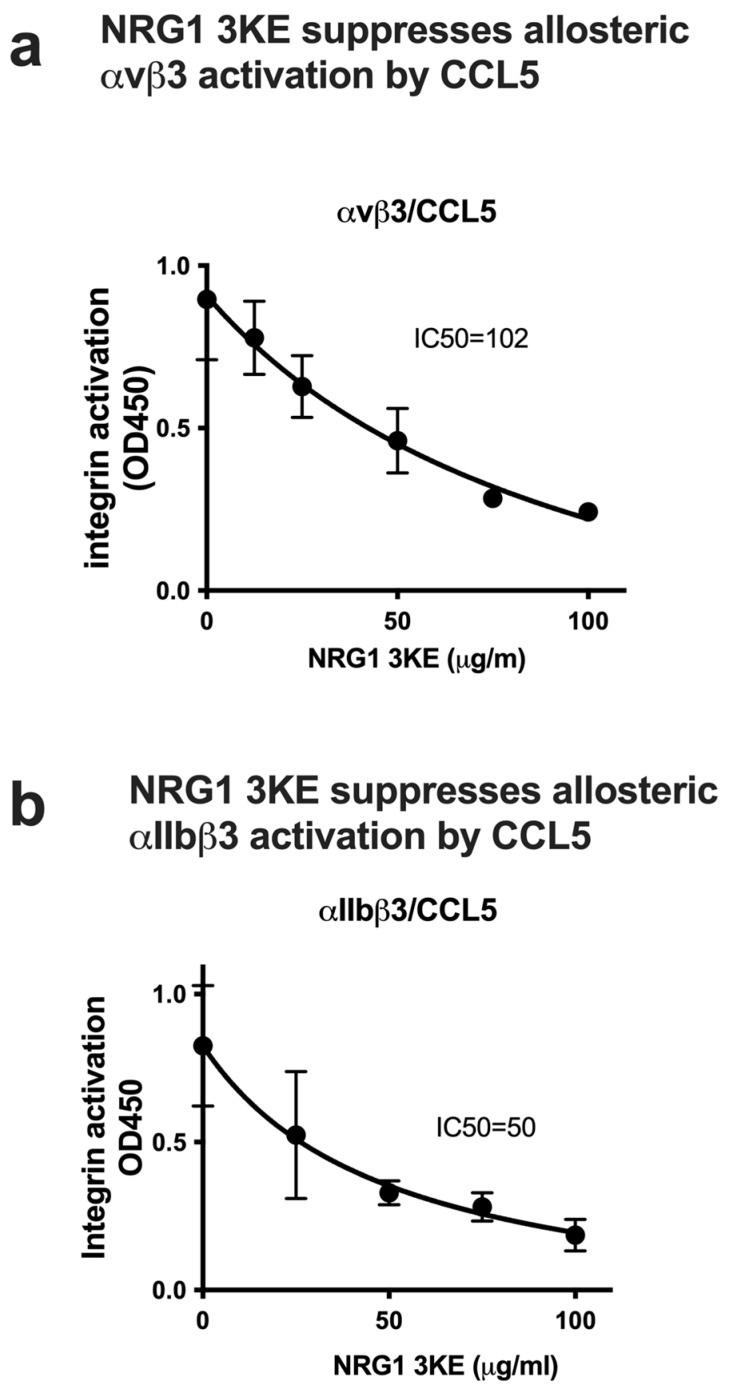
Non-mitogenic NRG1 3KE mutant suppressed allosteric activation of soluble αvβ3 (**a**) and αIIbβ3 (**b**). The 3KE mutant defective in binding to site 1 bound to site 2 and suppressed integrin activation by CCL5. Activation was assayed as described in Figure 4 or Figure 5. NRG1 suppressed integrin activation by CCL5 (at 6.25 μg/mL) in a dose-dependent manner in ELISA-type activation assays. Wells were coated with γC399tr, a specific ligand for αvβ3, and incubated with soluble αvβ3. Wells were coated with γC390-411, a specific ligand for αIIbβ3, and incubated with soluble αIIbβ3 (1 μg/mL). The 3KE mutant defective in binding to site 1 bound to site 2 and suppressed activation of integrins by CCL5. Background binding in the absence of cytokines was not subtracted. Data are shown as average ± SD in triplicate experiments. Prism 10 was used for statical analysis (n = 3).

**Figure 8 cells-14-00617-f008:**
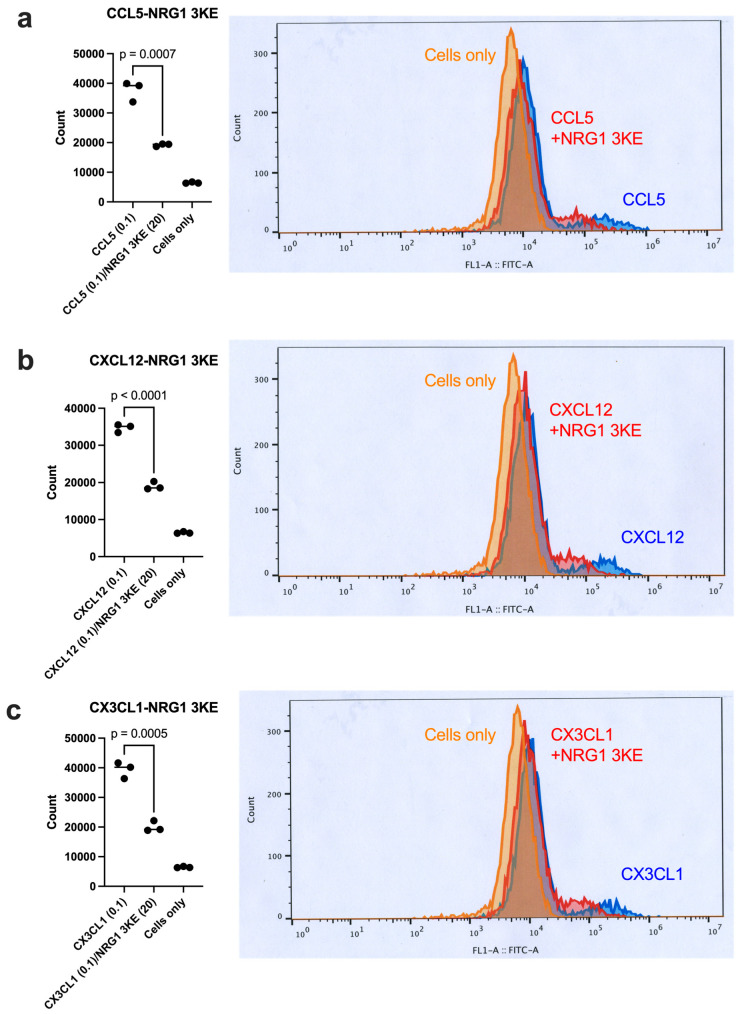
NRG1 3KE suppresses αIIbβ3 on the cell surface. The αIIbβ3-CHO cells were suspended in binding buffer I with 1 mM Ca^2+^ and incubated with inflammatory chemokines (**a**) CCL5, (**b**) CXCL12, and (**c**) CX3CL1 (0.1 μg/mL) with or without NRG1 3KE (20 μg/mL) and FITC-labeled γC390-411 fused to GST [18]. Integrin activation was detected by flow cytometry. Mean fluorescent intensity was used for evaluating integrin activation.

**Figure 9 cells-14-00617-f009:**
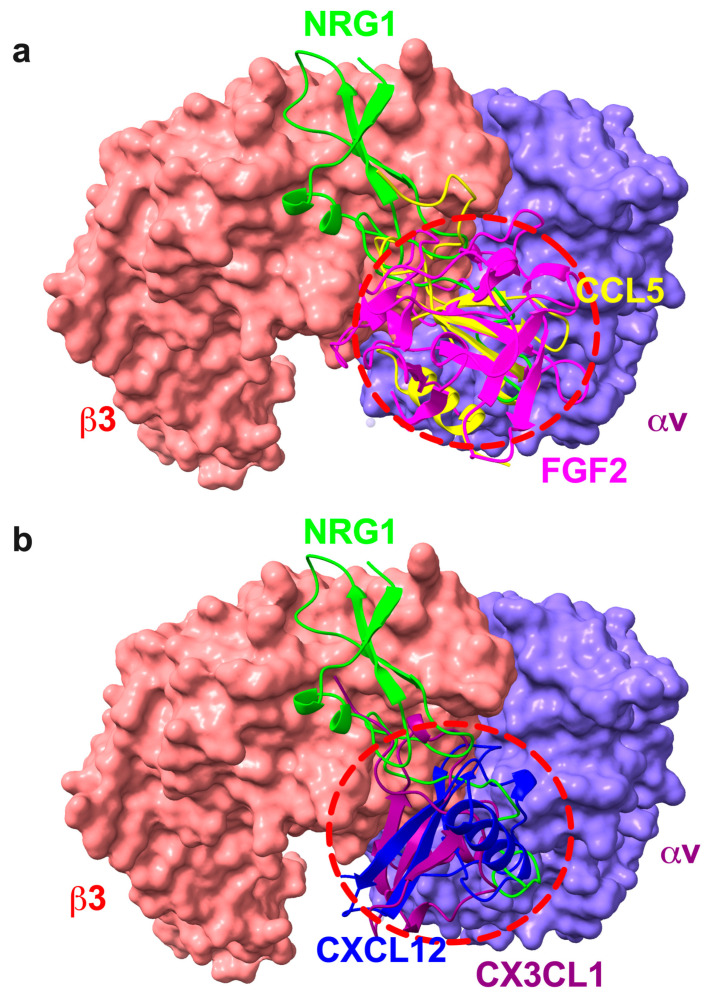
The predicted NRG1-binding site is off the center of site 2, where predicted binding sites for inflammatory cytokines are clustered. The chemokine domains of CX3CL1 [37], FGF2 [34], CXCL12 [16], CCL5 [18], and NRG1 EGF-like domain (the current study) have been predicted to bind to site 2 of αvβ3 (1JV2.pdb). The predicted binding sites for inflammatory cytokines are clustered in the center of site 2 (shown as red circles in (**a**,**b**)), which is consistent with the observation that they allosterically activated integrins. In contrast, the anti-inflammatory NRG1-binding site partly overlaps with those of inflammatory cytokines but mostly off the center of site 2. This is consistent with the observation that NRG1 did not activate integrins but inhibited integrin activation by the inflammatory cytokines.

**Table 1 cells-14-00617-t001:** Amino acid residues predicted to be involved in binding of NRG1 to site 2 of αvβ3.

NRG1 (1HAF.pdb)	αv (1JV2.pdb)	β3 (1JV2.pdb)
Leu3, Glu10, Phe13, Cys14, Met22, Val23, Ser27, Asn28, Pro29, Ser30, Arg31, Tyr32, Leu33, Cys34, Lys35, Cys36, Gln37, Pro38, Phe40, Thr41, Gly42, Ala43, Arg44, Cys45, Thr46, Glu47, Pro50, Val53, Asn55, Gln56	His91	Tyr122, Lys125, Leu128, Trp129, Gln132, **Pro160, Met165, Glu171, Glu174, Asn175, Pro176, Cys177, Lys181, Thr182, Thr183, Cys184, Leu185, Pro186, Met187, Phe188**, Lys191, Val207, Lys208, Lys209, Gln210, Ser211, Val212, Asp278, His280, Tyr281, Ser282, Thr285

Amino acid residues within 0.6 nm between NRG1 EGF and αvβ3 were selected using Pdb Viewer (version 4.1). Amino acid residues in β3 that were in the SDL loop are shown in bold, and those in the β3 site 2 peptide were underlined.

## Data Availability

Research data are available upon request.

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
