# Peer review of "Neuregulin-1 (NRG1) Binds to the Allosteric Binding Site (Site 2) and Suppresses Allosteric Integrin Activation by Inflammatory Cytokines: A Potential Mechanism of Anti-Inflammatory and Anti-Fibrosis Action of NRG1"

_cells, 2025, doi:10.3390/cells14080617_

Round 1
Reviewer 1 Report
Comments and Suggestions for Authors
This is a rather straightforward study by experienced groups that address how multiple inflammatory cytokines regulate cytokine signaling through integrin binding. Integrins have the classical-ligand binding site (site 1) and the allosteric binding site (site 2) for its ligands and are involved in cytokine signaling. The authors previously showed that multiple inflammatory cytokines bind to site 2, leading activation of integrins (14-18). They showed that neuregulin (NRG)1 binds to site 1 of integrin induces integrin-NRG1-ErbB3 ternary complex formation, which critical for its mitogenic action. The NRG1 mutant defective in integrin binding (3KE) is defective in inducing ErbB3 activation and acts as an antagonist for ErbB3 (9). They recently showed both FGF1 and FGF2 bound to site 2 but FGF1 suppresses integrin activation induced by FGF2 (24). Based on the finding that NRG1 has anti-inflammatory effects, they hypothesized that NRG1 binds to site 2 and blocks inflammatory signaling induced by inflammatory cytokines.
In this manuscript, they showed that docking simulation predicted that NRG1 binds to site 2, NRG1 competitively suppressed integrin activation by multiple pro-inflammatory cytokines. Furthermore, they showed that the non-mitogenic NRG1 3KE mutant still bound to site 2 and inhibited CCL5-induced integrin activation. Based on the above results they propose that site 2-mediated inflammatory signaling may be a common mechanism of anti-inflammatory action by NRG1 and FGF1 and NRG1 3KE may have potential as a therapeutic.
This work is a nice contribution to the field of the effect of NRG1 on Integrin activation by several inflammatory cytokines. All the figures are very nice graphic tools for readers’ understanding.
Author Response
See attached response to all reviewers. Reviewer 1 did not criticize the paper.
Rev 1
This is a rather straightforward study by experienced groups that address how multiple inflammatory cytokines regulate cytokine signaling through integrin binding. Integrins have the classical-ligand binding site (site 1) and the allosteric binding site (site 2) for its ligands and are involved in cytokine signaling. The authors previously showed that multiple inflammatory cytokines bind to site 2, leading activation of integrins (14-18). They showed that neuregulin (NRG)1 binds to site 1 of integrin induces integrin-NRG1-ErbB3 ternary complex formation, which critical for its mitogenic action. The NRG1 mutant defective in integrin binding (3KE) is defective in inducing ErbB3 activation and acts as an antagonist for ErbB3 (9). They recently showed both FGF1 and FGF2 bound to site 2 but FGF1 suppresses integrin activation induced by FGF2 (24). Based on the finding that NRG1 has anti-inflammatory effects, they hypothesized that NRG1 binds to site 2 and blocks inflammatory signaling induced by inflammatory cytokines.
In this manuscript, they showed that docking simulation predicted that NRG1 binds to site 2, NRG1 competitively suppressed integrin activation by multiple pro-inflammatory cytokines. Furthermore, they showed that the non-mitogenic NRG1 3KE mutant still bound to site 2 and inhibited CCL5-induced integrin activation. Based on the above results they propose that site 2-mediated inflammatory signaling may be a common mechanism of anti-inflammatory action by NRG1 and FGF1 and NRG1 3KE may have potential as a therapeutic.
This work is a nice contribution to the field of the effect of NRG1 on Integrin activation by several inflammatory cytokines. All the figures are very nice graphic tools for readers’ understanding.
Reviewer 2 Report
Comments and Suggestions for Authors
The manuscript presents evidence that Neuregulin-1 (NRG1) binds to the allosteric binding site (site 2) of integrins, as demonstrated using a plate-bound assay. The results suggest that this newly identified protein interaction may suppress integrin activation induced by inflammatory cytokines CCL5, CX3CL1, CXCL12, and FGF2. The experiments are well-designed, and the presentation is clear. However, a substantial portion of the related research has been conducted by the same research group.
My primary concern is that all the supporting evidence is derived from cell-free conditions using purified recombinant proteins. To strengthen the conclusions and align with the journal’s interests, at least some experimental evidence should be provided to demonstrate that this interaction occurs in a live system (either in cells or in an animal model). For instance, experiments confirming that NRG1 binds to integrins on the cell surface and competes with cytokine binding would significantly enhance the study’s impact.
Minor Concerns:
Additionally, some descriptions in the Materials and Methods section are missing important references or contain typographical errors. These should be carefully checked. Specific points include:
Missing reference: "His-tagged NRG1 was purified by Ni-NTA affinity chromatography from bacterial extracts under denaturing conditions in 8M urea. The purified His-tagged NRG1 preparations were refolded as described (Ref?)." Please provide the appropriate reference.
Incomplete strain information: "Proteins were synthesized in E. coli BL1 ?" The specific strain name appears to be incomplete. Should this be BL21?
Ambiguous notation: "We found that the cyclic site 2 peptide bound to CX3CL1 and sPLA2-IIA to a similar extent as the non-cyclized β3 site 2 peptides in ELISA-type assays. We designed the corresponding cyclic β1 peptide (C268-KLGGIVLPNDGQSHLENNMYTMSHYYC295, 28-mer cyclic β1 peptide), in which C281 is converted to S."
It is unclear what "S" refers to in this context. If it represents serine, it should be explicitly stated to avoid confusion.
Overall, while the study presents an interesting and potentially impactful discovery, addressing these concerns would strengthen its validity and clarity.
Author Response
Rev2
The manuscript presents evidence that Neuregulin-1 (NRG1) binds to the allosteric binding site (site 2) of integrins, as demonstrated using a plate-bound assay. The results suggest that this newly identified protein interaction may suppress integrin activation induced by inflammatory cytokines CCL5, CX3CL1, CXCL12, and FGF2. The experiments are well-designed, and the presentation is clear. However, a substantial portion of the related research has been conducted by the same research group.
My primary concern is that all the supporting evidence is derived from cell-free conditions using purified recombinant proteins. To strengthen the conclusions and align with the journal’s interests, at least some experimental evidence should be provided to demonstrate that this interaction occurs in a live system (either in cells or in an animal model). For instance, experiments confirming that NRG1 binds to integrins on the cell surface and competes with cytokine binding would significantly enhance the study’s impact.
Response: We added the data (Figure 7), in which NRG1 3KE bound to the cell surface integrin aIIbb3 and suppressed integrin activation induced by inflammatory cytokines. As the reviewer suggested this experiment enhanced the biological significance of our study. Thank you.
Minor Concerns:
Additionally, some descriptions in the Materials and Methods section are missing important references or contain typographical errors. These should be carefully checked. Specific points include:
Missing reference: "His-tagged NRG1 was purified by Ni-NTA affinity chromatography from bacterial extracts under denaturing conditions in 8M urea. The purified His-tagged NRG1 preparations were refolded as described (Ref?)." Please provide the appropriate reference.
Response: We included reference 7 for refolding.
Incomplete strain information: "Proteins were synthesized in E. coli BL1 ?" The specific strain name appears to be incomplete. Should this be BL21?
Response: It is BL21. Corrected. Thanks.
Ambiguous notation: "We found that the cyclic site 2 peptide bound to CX3CL1 and sPLA2-IIA to a similar extent as the non-cyclized β3 site 2 peptides in ELISA-type assays. We designed the corresponding cyclic β1 peptide (C268-KLGGIVLPNDGQSHLENNMYTMSHYYC295, 28-mer cyclic β1 peptide), in which C281 is converted to S."
It is unclear what "S" refers to in this context. If it represents serine, it should be explicitly stated to avoid confusion.
Response: “S” was changed to S281.
Overall, while the study presents an interesting and potentially impactful discovery, addressing these concerns would strengthen its validity and clarity.
Response: Thank you for the encouraging comment.
Reviewer 3 Report
Comments and Suggestions for Authors
This manuscript reports the possibility that Neuregulin-1 (NRG1), which is known to suppress the progression of inflammatory diseases, fibrosis, and insulin resistance, can act as an antagonist of site 2 on integrins (αvβ3 and αIIbβ3) and block their allosteric activation by multiple cytokines. Moreover, a non-mitogenic NRG1 3KE mutant was found to still bind to site 2 and inhibit CCL5-induced integrin activation, suggesting that NRG1 3KE may have potential as a therapeutic.
The authors find that NRG1 binds to isolated peptides bearing the sequences of β3 site 2 and the specificity-determining loop (SDL), and this finding was confirmed by docking simulations. However, the same group previously reported (ref. 9) that NRG1 binds to the RGD site of the αvβ3 integrin. I do not believe that binding to isolated peptides can demonstrate the same interaction in the full protein. Therefore, to move the research forward, I suggest performing displacement experiments on full integrins using ligands with known binding sites, such as RGD-containing peptides.
The main result of this work is that the non-mitogenic mutant of NRG1 (3KE) still binds to site 2 of integrins and inhibits CCL5-induced integrin activation. However, this finding contradicts the experimental result reported in ref. 9, where the same authors demonstrated that the NRG1-3KE mutant is no longer able to bind the αvβ3 integrin.
Finally, the presumed inhibition of integrin activation is based on an assay using the binding of the integrin to a truncated form of the fibrinogen γ-chain. Again, more robust cell-based assays of integrin activation are required to demonstrate this statement.
Author Response
This manuscript reports the possibility that Neuregulin-1 (NRG1), which is known to suppress the progression of inflammatory diseases, fibrosis, and insulin resistance, can act as an antagonist of site 2 on integrins (αvβ3 and αIIbβ3) and block their allosteric activation by multiple cytokines. Moreover, a non-mitogenic NRG1 3KE mutant was found to still bind to site 2 and inhibit CCL5-induced integrin activation, suggesting that NRG1 3KE may have potential as a therapeutic.
The authors find that NRG1 binds to isolated peptides bearing the sequences of β3 site 2 and the specificity-determining loop (SDL), and this finding was confirmed by docking simulations. However, the same group previously reported (ref. 9) that NRG1 binds to the RGD site of the αvβ3 integrin. I do not believe that binding to isolated peptides can demonstrate the same interaction in the full protein. Therefore, to move the research forward, I suggest performing displacement experiments on full integrins using ligands with known binding sites, such as RGD-containing peptides.
Response: In our previous study (ref 9, Ieguchi et al, 2010), the predicted NRG1 binding interface to site 1 (9 amino acid residues in beta 3) were listed (Ref. 9, Table 1). In the present manuscript, we listed amino acid residues involved in site 2 binding in beta3 (Table 1). When we compared Table 1 of ref.9 (for site 1) and Table 1 of this manuscript (for site 2), only a single amino acid residue (Glu171) is common to site 1 and site 2 and remaining 14 out of 15 amino acid residues are only involved in NRG1-binding interface binding in site 2. This suggests that SDL is almost completely part of site 2 and is potentially useful to identify site 2-binding proteins. (This statement was added to the text.)
RGD-containing peptides may not be adequate for our study, since RGD peptide may bind to both site 1 and site 2, since the two binding sites have overlapping ligand specificities. It would be difficult to determine if RGD-containing peptides block site 1 binding or site 2 binding. We hope to develop small molecules that bind only to site 2 in the future experiments.
The main result of this work is that the non-mitogenic mutant of NRG1 (3KE) still binds to site 2 of integrins and inhibits CCL5-induced integrin activation. However, this finding contradicts the experimental result reported in ref. 9, where the same authors demonstrated that the NRG1-3KE mutant is no longer able to bind the αvβ3 integrin.
Response: Previous work (ref. 9) focused on NRG1 binding to site 1 of avb3 in 1 mM Mn2+ (which activates integrins), which results in avb3-NRG1-ErbB3 ternary complex and potent mitogenicity. The 3KE mutation inhibits NRG1 binding to site 1 and inhibits mitogenicity. The present manuscript showed NRG1 and NRG1 3KE bind to site 2. Site 2 is open when integrins are not active (as in 1 mM Ca2+). The 3KE mutations do not block binding to site 2 (Figure 1a). We believe that the two studies do not contradict to each other.
Finally, the presumed inhibition of integrin activation is based on an assay using the binding of the integrin to a truncated form of the fibrinogen γ-chain. Again, more robust cell-based assays of integrin activation are required to demonstrate this statement.
Response: We added the cell-based assays (Figure 7), in which excess NRG1 3KE bound to the cell surface integrin aIIbb3 and suppressed integrin activation induced by inflammatory cytokines. As the reviewer suggested this experiment enhanced the biological significance of our study.
Round 2
Reviewer 2 Report
Comments and Suggestions for Authors
Thanks for the response. I don't have further question.
Author Response
The response to this reviewer is in the comments to the Editor. Briefly, we included new Fig. 3 to clarify the binding of NRG1 to site 1 and site 2. Also, we edited the text to modify the repetitive part.
Reviewer 3 Report
Comments and Suggestions for Authors
The manuscript has been updated with a new cell-based experiment. However, in my opinion, the results are still too limited to justify publication in the high-impact journal Cells.
Response: In our previous study (ref 9, Ieguchi et al, 2010), the predicted NRG1 binding interface to site 1 (9 amino acid residues in beta 3) were listed (Ref. 9, Table 1). In the present manuscript, we listed amino acid residues involved in site 2 binding in beta3 (Table 1). When we compared Table 1 of ref.9 (for site 1) and Table 1 of this manuscript (for site 2), only a single amino acid residue (Glu171) is common to site 1 and site 2 and remaining 14 out of 15 amino acid residues are only involved in NRG1-binding interface binding in site 2. This suggests that SDL is almost completely part of site 2 and is potentially useful to identify site 2-binding proteins. (This statement was added to the text.)
RGD-containing peptides may not be adequate for our study, since RGD peptide may bind to both site 1 and site 2, since the two binding sites have overlapping ligand specificities. It would be difficult to determine if RGD-containing peptides block site 1 binding or site 2 binding. We hope to develop small molecules that bind only to site 2 in the future experiments.
Reviewer reply: The authors should explain why the current docking results differ from those obtained in Ref. 9 using the same docking procedure.
Response: Previous work (ref. 9) focused on NRG1 binding to site 1 of avb3 in 1 mM Mn2+ (which activates integrins), which results in avb3-NRG1-ErbB3 ternary complex and potent mitogenicity. The 3KE mutation inhibits NRG1 binding to site 1 and inhibits mitogenicity. The present manuscript showed NRG1 and NRG1 3KE bind to site 2. Site 2 is open when integrins are not active (as in 1 mM Ca2+). The 3KE mutations do not block binding to site 2 (Figure 1a). We believe that the two studies do not contradict to each other.
Reviewer reply: The 3KE mutant does not bind to integrin αvβ3 at any site (neither Site 1 nor Site 2), according to the authors' previous experimental results (Ref. 9, page 31393): “…suggesting that the 3KE mutant is defective in binding to αvβ3. We obtained similar results using β3-CHO cells (data not shown). We were unable to determine a KD value for 3KE binding to integrin αvβ3 due to low binding in surface plasmon resonance (data not shown).”
Response: We added the cell-based assays (Figure 7), in which excess NRG1 3KE bound to the cell surface integrin aIIbb3 and suppressed integrin activation induced by inflammatory cytokines. As the reviewer suggested this experiment enhanced the biological significance of our study.
Since the 3KE mutant does not bind to the integrin, the underlying cause of the observed effect remains unclear.
Comments on the Quality of English Language
The English needs improvement throughout the manuscript.
Author Response
We inserted the new Fig. 3 to clarify the difference between site 1 and site 2. NRG1 binds to site 1 but NRG1 3KE does not, as we published in 2010 (ref. 9). NRG1 3KE is non-mitogenic but NRG1 3KE but binds to site 2.
Round 3
Reviewer 3 Report
Comments and Suggestions for Authors
In my opinion, the results are too limited to justify publication in the high-impact journal Cells.
Comments on the Quality of English Language
The English can be improved.
Author Response
We respectfully disagree to your comment. Our findings that NRG1 binds to site 2 and inhibits integrin activation (and subsequent inflammatory signals) will significantly influence the direction of the field (e.g., inflammation).